# The Identification of APOBEC3G as a Potential Prognostic Biomarker in Acute Myeloid Leukemia and a Possible Drug Target for Crotonoside

**DOI:** 10.3390/molecules27185804

**Published:** 2022-09-07

**Authors:** Chenchen Ma, Peng Liu, Siyuan Cui, Chang Gao, Xing Tan, Zhaopeng Liu, Ruirong Xu

**Affiliations:** 1Central Laboratory of Affiliated Hospital of Shandong University of Traditional Chinese Medicine, Jinan 250014, China; 2Department of Laboratory Medicine, The First Affiliated Hospital of Shandong First Medical University & Shandong Provincial Qianfoshan Hospital, Jinan 250014, China; 3Department of Hematology, Affiliated Hospital of Shandong University of Traditional Chinese Medicine, Jinan 250014, China; 4First Clinical Medical College, Shandong University of Traditional Chinese Medicine, Jinan 250014, China; 5Department of Medicinal Chemistry, Key Laboratory of Chemical Biology (Ministry of Education), School of Pharmaceutical Sciences, Shandong University, Jinan 250014, China; 6Institute of Hematology, Shandong University of Traditional Chinese Medicine, Jinan 250014, China

**Keywords:** AML, apolipoprotein B mRNA editing enzyme catalytic polypeptide-like 3G (APOBEC3G) gene, crotonoside, bioinformatics analysis, prognosis, anti-tumor activities

## Abstract

The apolipoprotein B mRNA editing enzyme catalytic subunit 3G (APOBEC3G) converts cytosine to uracil in DNA/RNA. Its role in resisting viral invasion has been well documented. However, its expression pattern and potential function in AML remain unclear. In this study, we carried out a bioinformatics analysis and revealed that the expression of APOBEC3G was significantly upregulated in AML, and high expression of APOBEC3G was significantly associated with short overall survival (OS). APOBEC3G expression was especially increased in non-M3AML, and correlated with the unfavorable cytogenetic risks. Additionally, Cox regression analyses indicated APOBEC3G is a hazard factor that cannot be ignored for OS of AML patients. In molecular docking simulations, the natural product crotonoside was found to interact well with APOBEC3G. The expression of APOBEC3G is the highest in KG-1 cells, and the treatment with crotonoside can reduce the expression of APOBEC3G. Crotonoside can inhibit the viability of different AML cells in vitro, arrest KG-1 and MV-4-11 cells in the S phase of the cell cycle and affect the expression of cycle-related proteins, and induce cell apoptosis. Therefore, APOBEC3G could be a potential drug target of crotonoside, and crotonoside can be considered as a lead compound for APOBEC3G inhibition in non-M3 AML.

## 1. Introduction

Acute myeloid leukemia (AML) is a heterogeneous hematology disease, which is characterized by the clonal expansion of abnormally differentiated blasts of myeloid lineage in the peripheral blood, bone marrow, and/or other tissues [1,2]. Patients with AML may present with symptomatic complications such as infection, bleeding, or anemia [3]. At present, AML is the most common acute leukemia in adults and accounts for about 1.3% of new cancer cases in the USA of the cases [4]. The fast development of AML, especially when rapid proliferation of malignant blasts is accompanied by tumor lysis syndrome or disseminated intravascular coagulation, is fatal [1]; therefore, it is meaningful to identify the potential biomarkers for AML treatment.

The apolipoprotein B mRNA editing enzyme catalytic polypeptide-like (APOBEC) family is a group of single-stranded DNA (ssDNA) cytidine deaminases that convert cytosine (C) to uracil (U) in DNA/RNA (Figure 1A) [5]. There are 11 members in the APOBEC family, including activation-induced cytidine deaminase (AID) and APOBEC1, 2, 3 (A/B/C/D/F/G/H), and 4 [6,7]. The APOBEC3 proteins play important roles in immunity through restricting viral infection (both DNA and RNA viruses) and retro-transposition, including the human immunodeficiency virus (HIV), hepatitis B virus (HBV), human T cell leukemia virus type 1 (HTLV1), human papilloma virus (HPV), adeno-associated virus (AAV), herpes simplex-1 (HSV-1), and Epstein−Barr virus (EBV) [5,8]. The mechanism is involved in the deamination of cytosine bases in nucleic acids and results in an alteration of the substrate DNA or RNA sequence in the DNA synthesis progress [7]. Specifically, the APOBEC3 subfamily can directly bind viral genomic RNA and sterically block the progression of reverse transcriptase and inhibit the synthesis of cDNA [9]. Among the APOBEC3 proteins, APOBEC3G shows higher-level expression than other APOBEC3s and contributes to its antiviral activity, and deamination-dependent restriction is likely the dominant mechanism for HIV-1 restriction [10,11]. Recently, APOBEC has emerged as a prominent putative enzymatic source of mutations in cancer [7]. The APOBEC3 subfamily has been found to greatly participate in protecting cells from endogenous and exogenous DNA-based pathogens [7,12]. Next-generation sequencing has revealed an APOBEC mutation signature in >50% of human cancer types, with a variable impact within each tumor ranging from <5% to >90% of all base substitution mutations [7]. APOBEC3s can catalyze the changes in the cancer genome to promote tumor evolution resulting in therapeutic resistance. The function of deamination provides a chronic source of DNA damage in cancers that could select TP53 inactivation, leading to some tumors evolving rapidly and manifesting heterogeneity [7]. Additionally, misregulated APOBEC3s contribute to somatic mutation in multiple human cancers. Thus, inhibiting these enzymes may prevent precancerous mutations from occurring, such as those involved in tumor recurrence, metastasis, and drug resistance. Among the APOBEC family, APOBEC3B and 3G were mostly reported. A number of cancers including breast [13], lung [14,15], gastric [16,17], ovarian [18], cervix [19], and head and neck [20] frequently display a mutation of APOBEC3B, which may be accompanied by other members. APOBEC3G was associated with liver [21] colorectal [22], prostate [23], pancreatic cancer [24], and myeloma [6,25]. To date, the expression profiles of APOBEC genes and their relationships with clinic pathological features and prognosis in leukemia are unclear. This study offers a new perspective on further exploring the potential prognostically and/or therapeutically effect of APOBEC targets in AML. 

Crotonoside (Figure 1B), a guanosine analog originally isolated from *Croton tiglium* L., is a key component in many traditional indications for the treatment of constipation, headache, abdominal and stomach pain, inflammation, and rheumatism [26]. Moreover, crotonoside was reported to selectively inhibit AML cell growth in vitro and suppressed AML tumor growth in vivo through the inhibition of FLT3 and HDAC3/6 [27].

In this paper, with the help of computer-aided drug design, it was found that crotonoside can bind well with APOBEC3G at the binding site. Therefore, we investigated the functions of crotonoside on AML cells and its possible inhibition activities on APOBEC3G.

## 2. Results

### 2.1. The Expression of APOBEC3B/G and the Correlations with Overall Survival of AML Patients

Since APOBEC3B and 3G are closely related to a variety of tumors. To confirm the relationships of APOBEC3B and 3G with AML, we first analyzed their mRNA expression in AML patients and normal groups. APOBEC3B/G expression data of 173 TCGA-LAML patients and 70 GTEx normal individuals were collected. The downloaded RNAseq data removed batch effects and standardization disposal and were then converted to TPM standardized data. As shown in Figure 2A,B, the mRNA level of APOBEC3G was significantly higher than that of its normal counterparts (*p* < 0.001), while the mRNA level of APOBEC3B was lower expressed in AML patients than in the normal group (*p* < 0.001). Then, Figure 2C,D were used to verify the expression difference in APOBEC3B/G further. The gene expression matrix was downloaded from another published GEO dataset (GSE63270), and the dataset was also standardized. 

Next, we analyzed the effect of APOBEC3B/G gene expression on the prognosis of the AML patients. The results indicated that the expression level of APOBEC3B has no direct relation to poor overall survival (OS) for patients with AML (HR (high) = 1.01; *p* = 0.981, Figure 2E); however, high APOBEC3G expression was considerably associated with the short OS of AML patients (Hazard ratio; HR) (high) = 1.89; *p* = 0.004, Figure 2F) compared with the low expression group. Furthermore, the time-dependent receiver operating characteristic (ROC) analysis on APOBEC3B/G was also performed, and the results showed APOBEC3B/G had moderate prediction abilities for 1 year (Figure 2G), while APOBEC3G had a larger area under the curve (AUC) (0.716) than APOBEC3B (0.538) for 3-year survival (Figure 2H), indicating that APOBEC3G had certain prediction abilities in predicting survival outcome in a long time. These results suggested that the unregulated expression of APOBEC3G may be a predictive biomarker of a poor OS for AML patients.

### 2.2. APOBEC3G Expression and Clinical Characteristics

To further illustrate the relationship between APOBC3G expression and clinical features, we explored the association between the mRNA expression of APOBEC3G and clinical laboratory parameters from TCGA datasets. Patients were divided into a high- and a low-expression group based on the median expression level of APOBEC3G as the optimal cut-off value by R software. As shown in Table 1, differences were not statistically significant between APOBEC3G high and APOBEC3G low AML patients according to gender, race, age, white blood cell (WBC) count, BM/PB blast, cytogenetic risks, and FLT3 and CEBPA mutations (*p* > 0.05 for all). However, there were statistical significances among different FAB types (*p* < 0.001) and NPM1 (wild type vs. mutation) (*p* = 0.018) expression level. According to FAB classification, AML-M3 is a subtype of AML characterized by the presence of promyelocytic leukemia-retinoic acid receptor alpha (PML-RARA) genes fusion. Among the patients with a high expression of APOBEC3G, M3 patients accounted for the smallest proportion (0%, 0/76). Moreover, among the patients in NPM1 wild type, the APOBEC3G high group had a higher frequency of (43.3%, 65/75) than patients in the APOBE3G low group (34.7%, 52/75), and the proportion of patients in the APOBEC3G mutation with a high expression of APOBEC3G is less (6.7%, 10/75, *p* = 0.018). Meanwhile, the mRNA expression levels of APOBEC3G according to different classifications of AML are shown in Figure 3. Among different FAB subtypes, APOBEC3G expression was lowest in M3 compared to other subtypes (Figure 3A). Additionally, in NPM1 wild type, the expression of APOBEC3G is lower than that of in mutation type (Figure 3B). These results were in accordance with the cytogenetic risks (the meaning of cytogenetic risks is illustrated in Appendix A), and in comparison with the favorable cytogenetic risk, the expression APOBEC3G was higher in the unfavorable group (Figure 3C). Meanwhile, it showed differences between subgroups with and without the other gene mutations such as FLT3, DNMT3A, KRAS, CEBPA, IDH1, IDH2, and RUNX1.

### 2.3. Univariate and Multivariate Cox Regression Analyses

We performed univariate Cox regression analyses to assess the factors influencing the OS. The age, cytogenetic risk, and gene mutations were treated as dichotomous or categorical variables, while the leukocyte count and blast cell percentage as continuous variables. The median expression level of each variable was adopted as the threshold value to dichotomize the cohort. As shown in Table 2, age (Hazard ratio = 3.333, *p* < 0.001), cytogenetic risk (Hazard ratio = 3.209, *p* < 0.001), FAB classification M5–M7 (Hazard ratio = 6.615, *p* < 0.001), and APOBEC3G expression (Hazard ratio = 1.893, *p* = 0.004) were identified as high-risk factors for poor OS. Next, old age (>60), patients with intermediate and poor cytogenetic risk, patients of M5 and M6 and M7 and a high expression of APOBEC3G were included in the multivariate Cox analysis. After adjustment for the other risk factors, a high expression of APOBEC3G (Hazard ratio = 1.971, *p* = 0.009) was identified as the second prognostic hazard factor after old age. According to the above regression analysis, we developed a nomogram containing our prognostic risk-scoring model and multiple clinical factors. In the TCGA cohort, age, cytogenetic risk, FAB classification, FLT3 and NPM1 mutations, and APOBEC3G were eventually selected to establish an accurate predictive nomogram. The nomogram model can be used to calculate the total points for risk stratification. Figure 4 reveals the crucial prognostic value of APOBEC3G for predicting the OS in AML.

### 2.4. Three APOBEC3G—Associated Pathways Were Enriched by GSEA

To explore the potential molecular mechanisms of APOBEC3G, GSEA was performed for high and low APOBC3G expression datasets to identify critical signaling pathways involved in AML. KRAS, IL-SAT5, and the Hedgehog signaling pathway were screened out based on APOBEC3G’s high expression, all of which were closely related to the development of cancer cells (Figure 5). The results showed that high APOBEC3G was most closely related to upregulating the KRAS signaling pathway, which played important roles in the cell cycle control of tumor initiation. However, no signaling pathway was enriched in an APOBEC3G low phenotype based on NES, with adjusted *p* value < 0.05 and FDR value < 0.05, indicating low expression of this gene was not significantly correlated with any pathway.

### 2.5. Docking Simulation of the Binding of Crotonoside with APOBEC3G

To investigate whether APOBEC3G would be a possible target for the natural product crotonoside, molecular docking simulations were carried out using the SYBYL-X suite. As a two-domain deaminase, APOBEC3G is composed of a pseudocatalytic domain (residues 1−196) and a catalytic domain (residues 197–384) [28], each domain has six α helices and five β strands. As shown in Figure 6, crotonoside can be well embedded in the binding pocket of APOBEC3G (PDB: 3V4K) and interacted GLU225, ARG278, LYS234, and GLU223 with hydrogen bonds at the center of the catalytic domain. Hydrogen bonds are also formed between crotonosides and conserved water molecules, which is helpful for binding mode. This docking simulation indicated that crotonoside has the potential to interact and inhibit the activity of APOBEC3G.

### 2.6. Crotonoside Can Inhibit the Viability of Different AML Cell Lines

We firstly evaluated the viability inhibitory activity of crotonoside on five different AML cell lines using the CCK-8 assay. The results indicated that crotonoside displayed moderate activities against these tested AML cells, and the IC_50_ values are shown in Figure 7. Crotonoside was found to significantly decrease the viability of AML cells in a concentration-dependent manner, and it was most sensitive to KG-1 with the IC_50_ value of 20.77 μM, whereas its inhibitory activity was not notable to THP-1 with an IC_50_ value of 60.69 μM. However, the activity of crotonoside on NB4 (the cell line was derived from a human acute promyelocytic leukemia (M3) patient) was not as good as that of KG-1 and MV-4-11.

### 2.7. Crotonoside Can Reverse the Highly Expressed ABOBEC3G in KG1 Cells

The endogenous mRNA expression levels of APOBEC3G in the four non-M3 AML cell lines were assayed by qPCR. The expression level was relatively higher in KG-1 cells than in others (Figure 8A). Next, we evaluated the expression of all the members of the APOBEC3 family in KG-1 cells with or without crotonoside. RNA transcriptome sequencing of KG-1 showed that the APOBEC3G was the most highly expressed and the APOBEC3D/F/G were relatively highly expressed, while the other members showed much lower expressions (Figure 8B). Exposing to crotonoside at the concentration of 20 μM, the high expression level of APOBEC3D/F/G was reversed dramatically. The qPCR result confirmed it again, when the KG-1 cells were treated with crotonoside at the concentration of 20 μM, the expression of APOBEC3G was decreased (Figure 8C). Therefore, crotonoside can reverse the high expression of APOBEC3G, and this result is in agreement with the molecular docking simulations, indicating that APOBEC3G may be a potential target of crotonoside.

### 2.8. Crotonoside Induced Cell Cycle Arrest and Inhibited the Expression of the Related Proteins

To examine whether crotonoside inhibits cell viability via cell cycle arrest, we performed flow cytometry analysis to investigate the cell cycle distribution. Both KG-1 and MV-4-11 cells were treated with different concentrations of crotonoside for 48 h. Cell cycle progression was analyzed, and the obtained results revealed that the cells were arrested at the S phase in a dose-dependent manner. As shown in Figure 9A, the proportion of cells in the S phase reached 42.29% for KG-1 cells when the cells were treated with 20 μM crotonoside, whereas the proportion of cells in the S phase was only 23.40% for the control. The same performance was shown in MV-4-11 (Figure 9B). The proportion of MV-4-11 cells in the S phase gradually increased with the increase in crotonoside concentration, and reached 45.57% when incubated with 30 μM crotonoside, while the proportion was only 25.61% for untreated cells. We further detected the effects of crotonoside on cell cycle-related proteins cyclinB1, cyclinD1, CDK2, and C-myc. The crotonoside significantly decreased the protein expression of CDK6, CylinB1, CDK2, and C-myc in both KG-1 and MV-4-11 cells in a dose-dependent manner after treatment for 48 h at a different dose (Figure 9D).

### 2.9. Crotonoside Can Induce the Apoptosis of KG-1 and MV-4-11 Cells

To investigate the effects of crotonoside on cell apoptosis, both KG-1 and MV-4-11 cells treated with different concentrations of crotonoside were stained with annexin V-PE and 7-Aminoactinomycin D (7-AAD) and analyzed by flow cytometry. Dual staining for annexin V-PE and 7-ADD permitted the discrimination among live cells (annexin V-PE−/7-AAD−), early apoptotic cells (annexin V-PE+/7-AAD−), late apoptotic cells (annexin VPE+/7-AAD+), and necrotic cells (annexin V-PE−/7-AAD+). As exhibited in Figure 10A, after the KG-1 cells were exposed to 5, 10, and 20 μM of crotonoside for 48 hours, the total fractions of early apoptotic cells and late apoptotic cells were 7.12%, 11.92%, and 36.14%, respectively. In contrast, only 6.40% were found in the untreated control. Similarly, when the MV-4-11 cells were treated with crotonoside at 7.5, 15, and 30 μM of for 48 h, the total fractions of early apoptotic cells and late apoptotic cells were 30.90%, 39.51%, and 62.66%, respectively, while only 13.91% for the control (Figure 10B). The results confirmed that crotonoside significantly induced AML cell apoptosis in a dose-dependent manner.

## 3. Discussion

Gene mutations and chromosomal variations are the main factors in the pathogenesis of AML, and cytogenetic abnormality or aberrant gene expression was closely associated with prognosis in AML [29], which inspired us to explore new biomarkers for AML.

As an important member of the APOBEC family, APOBEC3G plays a pivotal role in liver, colorectal, prostate, pancreatic cancer, and myeloma by affecting single-stranded DNA (ssDNA) that is exposed during replication, transcription, or during DNA damage repair [25]. However, the expression profile of APOBEC3G in AML and the effect on the pathogenesis as well as the prognosis of AML remain obscure. In this study, we carried out a bioinformatics analysis and the results revealed that the APOBEC3G may be a potential prognosis-related biomarker for non-M3 AML. 

Based on the bio-information analysis, the transcriptional level of APOBEC3G was dramatically increased in AML compared to the control cohort, which is contrary to another tumor-related gene member APOBEC3B [30]. The prognostic values of APOBEC3B/G genes in patients with AML were assessed. Kaplan–Meier analysis suggested that APOBEC3B has no directive relationship with the OS of AML patients, whereas the APOBEC3G overexpression presented significantly shorter OS, and the ROC analysis showed APOBEC3G had a larger AUC of survival time than APOBEC3B for a 3-year prediction, which indicated that high APOBEC3G expression can be an unfavorable indicator for AML OS prediction. 

We examined the relationship between APOBEC3G expression and clinical features and genetic alterations of AML patients. A higher APOBEC3G expression was closely associated with special clinical features, including NPM1 mutation and FAB classifications. By studying the association between APOBEC3G expression and clinic pathologic features in AML, we found that the patients with APOBC3G high expression were more in the wild-type NPM1 group than in the mutation group. The RNA expression results also confirm that APOBEC3G expression was higher in the NPM1 wild-type group than in the mutation group. NPM1 mutation status accompanied by a mutation in a variety of other genes might be an indicator of favorable survival for AML [31]. However, in NPM1 wildtype (WT) AML patients, no reliable biomarkers have been identified. Wild-type NPM1 without FLT3-ITD or with FLT3-ITD low belongs to the cytogenetic intermediate risk group, while the wild-type NPM1 with FLT3-ITD high belongs cytogenetic poor group [32]. Correspondingly, the RNA expression results also showed it was higher in the cytogenetic intermediate and poor risk group compared with that of the favorable group. These results suggest the close relationships between the expression of APOBEC3G and the cytogenetic risk. FAB typing is another indicator with a significant difference. The cytogenetic of leukemic cells is the most relevant prognostic parameter for non-M3 AML [33]. Although the diagnosis and treatment of AML have made some progress, non-M3 AML patients are affected by many factors, and it is easy to relapse after the complete remission of chemotherapy, so it is difficult to obtain a good long-term prognosis. In this study, M3 patients with APOBEC3G high expression only account for a very small number of overall cases. The expression of APOBE3G in M3 patients was significantly lower than in that of non-M3 AML. These results indicated that APOBEC3G could be a new potential biomarker for comprehending the molecular basis of non-M3 AML and a potential target for drug developments 

To validate whether overexpression of APOBEC3G could be used as a tool for risk stratification, univariate and multivariate Cox regression analyses were performed, and the AML-related factors such as age, gender, WBC count, BM/PB blast count, cytogenetic risk, FAB classification, and FLT3 and NPM1, which are easily mutated genes, were evaluated [2,4]. The results showed that high APOBEC3G expression is the second highest risk for survival after old age. Based on Cox regression analyses, a nomogram was supplied containing a risk-scoring model, which is helpful for judging the prognosis of patients and guiding the treatment.

GSEA analysis gained further insight into the biologic pathways, and the KRAS, IL-SAT5, and hedgehog pathways were found to be closely relevant to APOBEC3G overexpression, with KRAS being the most involved hallmarks in AML. KRAS mutation was observed with very high frequencies in multiple KRAS-driven tumors, such as pancreatic, colorectal, and non-small cell lung carcinoma [34]. KRAS signaling can affect tumor proliferation by two independent pathways: one is the MAPK/ERK and PI3K/AKT signaling pathway and the other is YAP1 and c-myc, both of which were involved in cell cycle actions [34]. However, no signaling pathways were enriched in the APOBEC3G low phenotype. The regulatory effect of this gene on the KRAS pathway may be interesting and the focus of our future investigations.

Due to the importance of APOBEC3G to AML, it is of interest to find new therapeutic agents that can inhibit APOBEC3G expression. The conventional chemotherapies for AML patients were limited by the many side effects such as hepatotoxicity, myelosuppression, and tumor lysis syndrome [35]. Traditional Chinese Medicine (TCM) has been developed for over 2500 years [36], and a variety of natural product medicines have exhibited excellent anti-tumor activities with different mechanisms. We have been devoted to screening bioactive compounds that have the potential for AML treatment in recent years, and virtual drug screening had been conducted to find new drugs that can inhibit APOBEC3G. Crotonoside, the isomer of guanosine, is a natural product extracted from the Chinese medicinal herb *Croton tiglium* L. It was reported to suppress the growth of the AML cell MV-411 by inhibiting FLT3 and HDAC3/6 [27]. A molecular docking simulation indicated that crotonoside can bind with APOBEC3G, suggesting that APOBEC3G can be a potential target of crotonoside. 

In this study, we evaluated the viability inhibitory activities of crotonoside against five different AML cell lines. Crotonoside significantly decreased AML cell viability in a concentration-dependent manner, with KG-1 being the most sensitive cell. The mRNA level of APOBEC3G was higher in KG-1 cells than that of other tested cells. To confirm this, RNA sequencing of the APOBEC family was conducted in KG-1 cells. APOBEC3G was the highest expressed gene, and the APOBEC3D/F/G was relatively highly expressed compared to others. By exposure to crotonoside, the expressions of APOBEC3D/F/G were reversed, while the other members were little affected. It can be inferred that crotonoside had certain selectivity toward APOBEC3G. The qPCR verified our hypothesis that crotonoside could inhibit KG-1 cells by inhibiting APOBEC3G expression significantly. 

Cell cycle deregulation is a strategy of cancer therapy. Does the gene affect DNA synthesis by interfering with DNA bases, thereby affecting the cell cycle? To further confirm our prediction, we evaluated the cell cycle arrest activity of crotonoside against KG-1 and MV-4-11 cells by flow cytometry. Crotonoside was found to induce S phase arrest dramatically in a dose-depended manner, especially at the concentration of IC_50,_ which indicated that the cells were arrested at the DNA synthesis phase. Because APOBEC3G can lead to a change in the DNA base, inhibiting this gene leads to a large number of cells stagnating in the stage of DNA synthesis, while the cells in the G2 phase are significantly reduced. Whether this gene is related to the checkpoint from the S phase to the G2 phase is the key to our next exploration. CDK2 is crucial to the S-phase of the cell cycle [37]. Dose crotonoside inhibit cell cycle-associated proteins by inhibiting APOBEC3G to affect DNA synthesis? Western blot results verified our hypothesis. The expression of CDK2 was significantly inhibited with the increase in the concentration of crotonoside. Overexpression of CDK4/6 leads to the activation of the anti-aging mechanism of tumor cells and promotes the formation of tumors. Overexpression of CDK6 leads to enhancement of the function of the cell cycle regulator FoxM1 and promotes the cell cycle from the G1 phase to the S phase [38]. In addition, overexpression of CDK6 can also promote the expression of inflammatory mediators by regulating NF-κB to form tumors. Moreover, CDK6 is a direct target of the MLL-AF9 fusion protein, which can promote the proliferation of AML cells [39]. The expression of CDK6 in AML cells decreased significantly after treatment with crotonoside. Since the activation of CyclinB-CDK1 is generally considered to trigger the entry to mitosis [40]. CyclinB1 was evidently degraded and almost undetectable when the cells were treated with crotonoside at a high concentration. c-Myc, another key element in controlling cell cycle, differentiation, and apoptosis [41,42,43], has also been demonstrated to be down-regulated after the treatment with different concentrations of crotonoside. 

We also explore the induction of apoptosis by crotonoside. Flow cytometry analysis revealed that the apoptotic rate of KG-1 cells increased significantly in a dose-dependent manner from 6.40% for control to 36.14% at 20 μM of crotonoside. Similar performance was obtained in MV-4-11 cells, the apoptotic rate increased from 13.91% for control to 62.66%. These results were in accordance with the CCK-8 assay and confirmed the cytotoxicity of crotonoside.

## 4. Materials and Methods

### 4.1. RNA-Sequencing Datasets Acquisition and Processing

Clinical data and RNA sequences of a total of 173 AML patients were obtained from the TCGA (The Cancer Genome Atlas) database (https://portal.gdc.cancer.gov/repository; accessed on: 10 July 2021), which is referred to as the TCGA-LAML (Leukemia Acute Myeloid Leukemia) cohort in this study and 151 patients’ peripheral blood data were eligible for inclusion by removing patients without clinical data. The sample data of healthy individuals (whole blood) as the control group were downloaded from the GTEx (Genotype-Tissue Expression) database (https://gtexportal.org/; accessed on: 10 July 2021) [44,45,46]. The RNA-seq data of TCGA and GTEx were downloaded from UCSC XENA (https://xenabrowser.net/datapages/; accessed on: 10 July 2021) in TPM format, which was uniformly processed in advance through the toil process, eliminating the batch effect [47,48]. The AML expression profiling dataset GSE63270 (*n* = 104) was downloaded from the GEO (https://www.ncbi.nlm.nih.gov/geo/; accessed on: 10 July 2021) (National Center for Biotechnology Information (NCBI), USA) [49], which includes expression data for healthy and AML bone marrow samples.

### 4.2. Statistical Analyses

Both the log2 (TPM+1) values of RNA sequencing data provided by TCGA and GTEx and the log2 (FPKM) values of the RNA sequencing data from GSE63270 were compared by Shapiro–Wilk normality test, which were described as medians and interquartile range (IQR) and examined by the Mann–Whitney U test. GraphPad Prism 8.0.2 (San Diego, CA, USA) was used to draw the dot plot. 

The TCGA-LAML cohort was dichotomized into higher or lower expression groups using the median APOBEC3G expression value of the cohort as a cut-off. Overall survival (OS) was defined as the time from the initial diagnosis to death for any reason or the end of observation. The OS prediction was performed by the “survival package‘’ of R software (The University of Auckland, Auckland, New Zealand). Time-dependent receiver operating characteristic (ROC) curve was performed to evaluate the predictive power of the gene profile, followed by calculation of the area under the curve (AUC), with the “time ROC package” of R software. 

Gene expression correlation analyses and statistics were completed using R3.6.3. The relationships between gene expression and clinical features were determined by Student’s *t*-test or One-way ANOVA test and visualized by GraphPad prism 8.0.2.

Univariate and multivariate Cox regression analyses were performed to identify candidate prognostic genes in the LAML cohort from TCGA. The cohort contains 140 AML patients with high-throughput sequencing (RNA-Seq) data. The 3 samples missing detailed clinical information and the number of samples included was 137. The *p*-value, HR, and 95% CI of each variable were built through “survival” and “survminer” R packages in RStudio 4.1.2. Gene expression levels were dichotomized based on the median expression level in the cohort as the cut-off value.

Nomogram was established by “rms” and ‘’survival” R packages in RStudio 4.1.2. The TCGA-LAML cohort contains 140 AML patients, 7 samples with missing variable information, and the number of samples finally included was 133. 

Analyses of statistical significance for differences between the control and treated groups were determined by Student’s *t*-test or One-way ANOVA test and visualized by GraphPad prism 8.0.2. The comparative cytological experiment data were expressed as the mean ± standard deviation of at least 3 independent experiments. *p* < 0.05 was considered statistically significant.

### 4.3. Gene Set Enrichment Analysis

Gene Set Enrichment Analysis (GSEA) is a computational method that determines whether a priori-defined set of genes shows statistically significant [50]. GSEA software version 4.1.0 (Cambridge, MA. USA) was used for this study. The 151 patients from the TCGA-LAML cohort were divided into the APOBEC3G high group and APOBEC3G low group based on the median value of APOBEC3G expression. All the RNAseq expression matrix of these patients was used as the input file for the GSEA analysis. The hallmark gene sets database v7.5.1 was selected, and the number of permutations was set as 1000. The normalized enrichment score (NES), nominal *p* value (NOM *p*), and false discovery rate q value (FDR q) were selected to classify enriched signal pathways. Gene sets with |NES| > 1, NOM *p* < 0.05, and FDR q < 0.25 were regarded as significantly enriched.

### 4.4. Molecular Modeling

The X-ray crystal structure of APOBEC3G was obtained from the RCSB Protein Data Bank (PDB codes: 3V4K). The protein structure was prepared with the SYBYL-X suite (version 1.3, Tripos, Princeton, NJ. USA). Hydrogen atoms were added to the crystal and charges were added to the biopolymer by AMBER7 FF99 force field, and the protocol was generated. The 3D structures of crotonoside were prepared by Chem3D Pro 14.0 (Cambridge, MA, USA) and optimized by the Tripos force field of SYBYL-X suite. The Surflex-dock module was used for the docking studies and the related parameters implied in the program were kept at default. The conformations were used to analyze the interactions between ligand and APOBEC3G. The docking result was visualized by PyMol (New York, NY. USA).

### 4.5. Cell Culture

KG-1 and Kasumi and NB4 cells (The Cell Bank of Chinese Academy of Sciences, Shanghai, China) were cultured in IMDM medium MacGene (M&C GENE TECHNOLOGY LTD., Beijing, China) supplemented with 20% fetal bovine serum (Gibco; Grand Island, NY, USA) and 1% penicillin/streptomycin; MV-4-11 cells were cultured in IMDM medium supplemented with 10% fetal bovine serum (Gibco) and 1% penicillin/streptomycin; NB4 and THP-1 cells (The Cell Bank of Chinese Academy of Sciences) were cultured in Roswell Park Memorial Institute 1640 medium (HyClone, Logan, UT, USA) supplemented with 10% fetal bovine serum (Gibco) and 1% penicillin/streptomycin. The cells were maintained in a humidified incubator with 5% CO_2_ at 37 °C.

### 4.6. Cell Viability Assay

Cell viability was assessed using a cell counting kit-8 (CCK-8) assay according to the manufacturer’s protocol (Beyotime, Shanghai, China). The doubling times of KG-1, MV-4-11, NB4, and THP-1 cell lines were 20–30 h [51], and the doubling time of Kasumi-1 was 40–45 h under experimental conditions [52]. Crotonoside (National Institutes for Food and Drug Control, Beijing, China) was dissolved in DMSO and diluted in different cell culture mediums to concentrations (100, 50, 25, 12.5, 6.25 μM), making sure the volume of DMSO was less than 0.1% of the culture medium volume, The exponentially growing cells were seeded into 96-well cell plates at 5–8 × 10^3^ cells/well, and incubated with gradient concentrations of crotonoside, with 0.1% DMSO solution being used as the control vehicle. Then, the cells were incubated at 37 °C for 48 h in a humidified chamber containing 5% CO_2_. CCK-8 solution (10 μL) was added to each well, and the plates were incubated for 1 h at 37 °C. The absorbance of the cells at 450 nm (OD450) was measured in a microplate reader (Thermo Fisher Scientific, Waltham, MA, USA). The half-maximal inhibitory concentration (IC_50_) was calculated based on the relative survival curve using GraphPad prism 8.0.2.

### 4.7. Cell Cycle Analysis 

KG-1 and MV-4-11 cells were seeded in 6-well plates at a density of 5 × 10^5^ cells per well, respectively, and treated with different concentrations of crotonoside for 48 h, with 0.1% DMSO solution being used as the control vehicle. Then, cells were harvested and washed by the ice-cold PBS. The cell cycle was assessed using a GENMED Universal periodic flow cytometry kit according to the manufacturer’s protocol (GENMED SCIENTIFICS, Shanghai, China). The samples were detected by a flow cytometer (Beckman Coulter Navios, Brea, CA, USA).

### 4.8. Apoptosis Analysis

KG-1 and MV-4-11 cells were seeded in 6-well plates at a density of 5 × 10^5^ cells per well, and treated with different concentrations of crotonoside for 48 h, with 0.1% DMSO solution being used as the control vehicle. Then, cells were harvested and washed by the ice-cold PBS. After suspension by Annexin V binding buffer, the cells were stained by Annexin V-PE and 7-AAD by using a Vazyme apoptosis detection kit according to the manufacturer s protocol (Vazyme, Nanjing, China). The samples were detected by a flow cytometer (Beckman Coulter Navios, Brea, CA, USA).

### 4.9. Western Blotting

Six-well plates were seeded with KG-1 or MV-4-11 cells at 5 × 10^5^ cells per well. Cells were first incubated with different concentrations of crotonoside (0, 5, 10, 5, and 20 μM) for 48 h, and a 0.1% DMSO solution was used as the control vehicle. The cells were harvested and lysed using RIPA lysis buffer. The protein concentrations were confirmed by a BCA Protein Assay Kit (Beyotime) (Shanghai, China). Protein samples were fractionated using 10% polyacrylamide gels and transferred onto PVDF membranes, then blocked by 5% nonfat milk for 1 h. Blots were incubated with primary antibodies against APOBEC3G (Abcam, Cambridge, UK), CyclinB1, CDK2, CDK6, CyclinD1, and c-Myc (Cell Signaling Technology, Danvers, MA, USA) overnight at 4 °C, followed by incubation with appropriate peroxidase-conjugated secondary antibodies. β-actin (Cell Signaling Technology) or GADPH (Santa Cruz Biotechnology, Santa Cruz, CA, USA) served as an internal control. The immunocomplexes were visualized by using an enhanced chemiluminescence detection system (Millipore, Burlington, MA, USA), followed by exposing the blots to X-ray film.

### 4.10. Quantitative Polymerase Chain Reaction Analysis

Total RNA was prepared from AML cells by using a SPARKeasy Improved Tissue/Cell RNA kit (SparkJade, Jinan, China) according to the manufacturer’s protocol. RNA yield was determined using a NanoDrop 2000 spectrophotometer (Thermo Fisher Scientific, Wilmington, DE, USA). Complementary DNA was synthesized using a SPARKscript II RT Plus Kit (SparkJade, Jinan, China). Quantitative polymerase chain reaction (qPCR) assays were performed using a 2× SYBR Green qPCR Mix (SparkJade, Jinan, China) in a 20 μL reaction volume, by Roche QRT-PCR System according to the manufacturer’s protocol. The messenger RNA (mRNA) levels of target genes were normalized to the mRNA level of β-actin. The sequences of the primers used for the APOBEC3G were: forward, 5’-TTGCCCGCCTCTACTACTTCTGG-3’ and the reverse, 5’-CTTGCTCCAACAGTGCTGAAATTCG-3’ (Sangon Biotech, Shanghai, China). The cycling conditions used were: initial denaturation at 94 °C for 3 min, followed by 40 cycles at 94 °C for 10 s, 60 °C for 30 s.

## 5. Conclusions

In conclusion, in this study, we identified that APOBEC3G could be a potential prognostic marker of AML. The detection of cycle and apoptosis-related proteins once again confirmed that crotonoside can inhibit the high expression of APOBEC3G, which is a possible target of crotonoside. Moreover, crotonoside can be considered as a potential leading compound for non-M3AML. Since the biological activity of crotonoside still needs to be improved, structure modifications are expected to find new candidate drugs for AML.

## Figures and Tables

**Figure 1 molecules-27-05804-f001:**
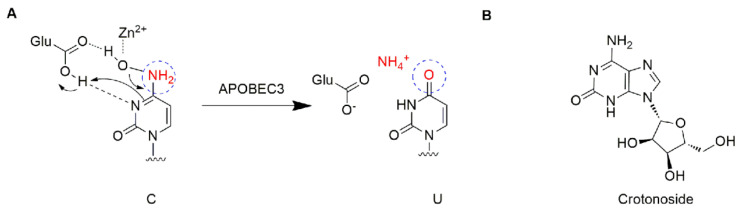
Structure description: (**A**) The proposed mechanism of ssDNA C-to-U deamination mediated by APOBEC3. (**B**) The structure of crotonoside.

**Figure 2 molecules-27-05804-f002:**
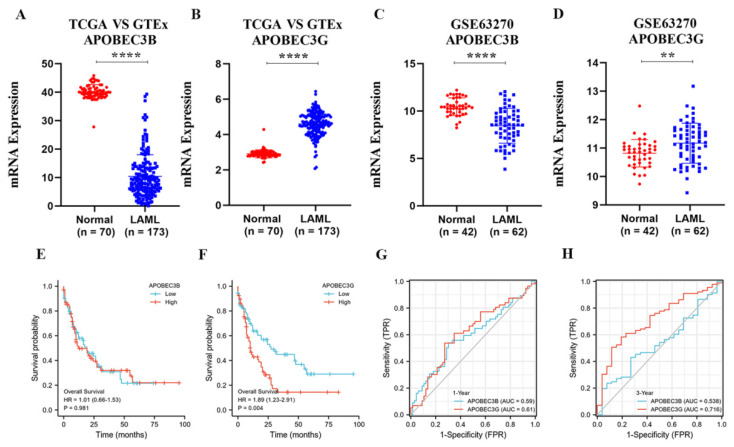
The expression of APOBEC3B/G and the correlations with overall survival of AML: (**A**,**B**) Dot plot show the expression profiles of APOBEC3B/G of the AML patients in the TCGA-LAML cohort (*n* = 173) compared to normal control (*n* = 70) from GTEx; **** *p* < 0.0001 vs. normal group. (**C**,**D**) Dot plot show the expression profiles of APOBEC3B/G of AML patients (*n* = 42) compared to normal control (*n* = 62) in GSE63270 cohort; **** *p* < 0.0001, ** *p* < 0.01 vs. normal group. (**E**,**F**) Prognosis analysis on OS (Overall Survival) was conducted based on the expression of APOBEC3B/G and survival status in the TCGA-LAML cohort (*n* = 140 after case-wise deletion) by R studio. Patients were dichotomized into a high- and a low-expression group, with the median mRNA expression level as the cutoff value, *p* value, and HRs (Hazard Ratio) were computed by log-rank test and Cox regression to draw the KM (Kaplan–Meier) curves; (**G**,**H**) The predictive accuracy of APOBEC3B/G were evaluated by time-dependent ROC (receiver operating characteristic) analysis at 1 or 3 years. AUC (area under the curve) values present the prediction ability.

**Figure 3 molecules-27-05804-f003:**
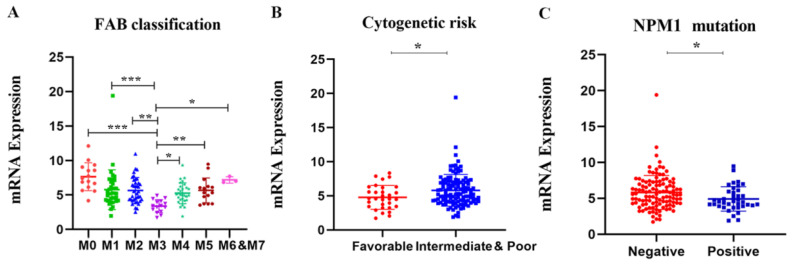
The mRNA expression levels of the APOBEC3G in various classes of AML in the TCGA database. Individual value plots representing APOBEC3G expression levels (**A**) in different FAB subtypes, * *p* < 0.05, ** *p* < 0.01, *** *p* < 0.001 vs. M3; (**B**) in different cytogenetic risks, * *p* < 0.05 vs. favorable group; (**C**) with NPM1 mutation versus wild type, * *p* < 0.05 vs. negative group.

**Figure 4 molecules-27-05804-f004:**
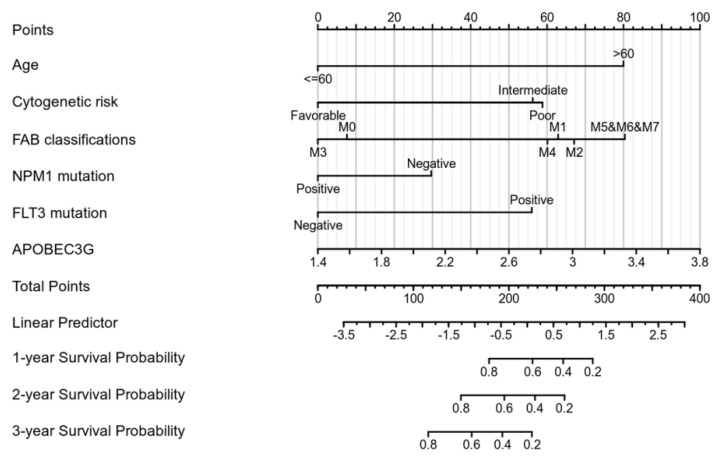
Nomogram for predicting the probability of 1-, 2-, and 3-year OS for AML.

**Figure 5 molecules-27-05804-f005:**
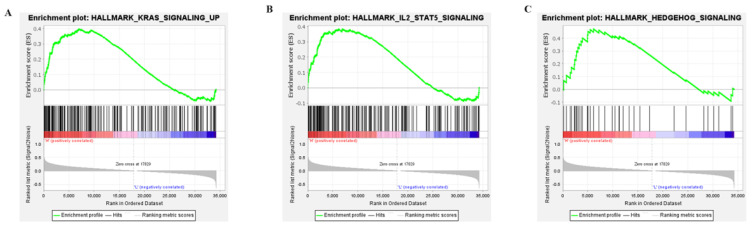
Enrichment plots from the gene set enrichment analysis (GSEA): (**A**) KRAS signaling; (**B**) IL2-STAT5 signaling; (**C**) HEDGEHOG signaling.

**Figure 6 molecules-27-05804-f006:**
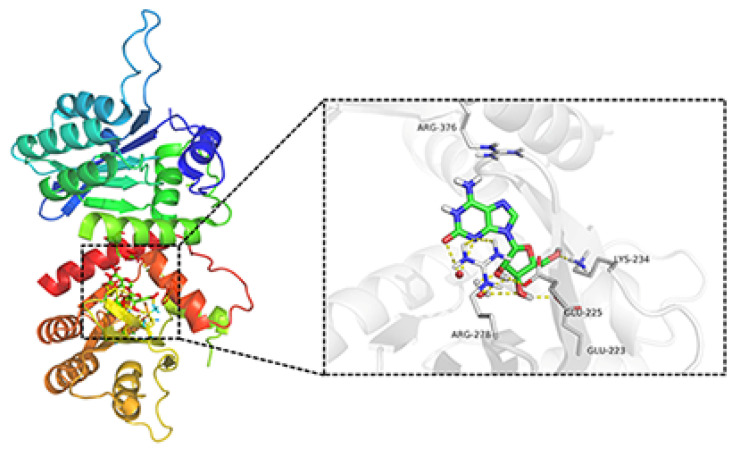
Molecular docking studies of crotonoside and APOBEC3G (PDB: 3V4K) at the binding site.

**Figure 7 molecules-27-05804-f007:**
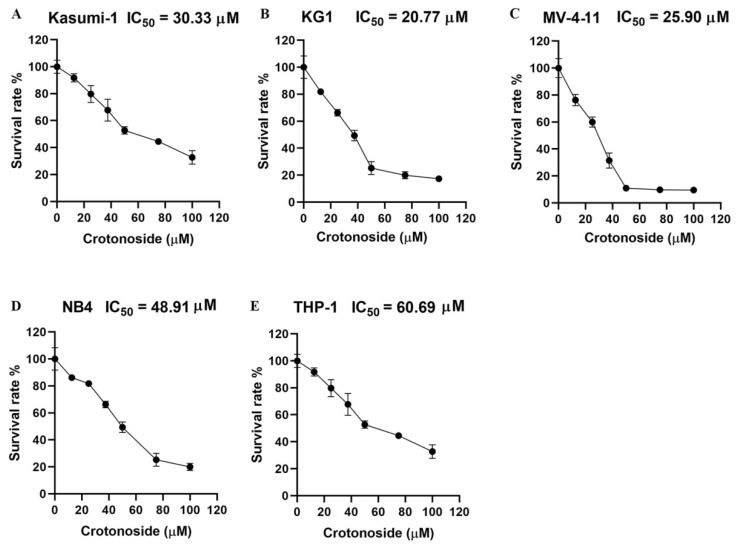
The viability inhibitory activity of crotonoside against different AML cells: (**A**) Kasumi-1, (**B**) KG-1, (**C**) MV-4-11, (**D**) NB4, and (**E**) THP-1 cells were treated with different concentrations of crotonoside for 48 h, and 0.1% DMSO was used as control. The cell viability was inhibited in a dose-dependent manner. All the data are presented as means ± SD from three independent experiments.

**Figure 8 molecules-27-05804-f008:**
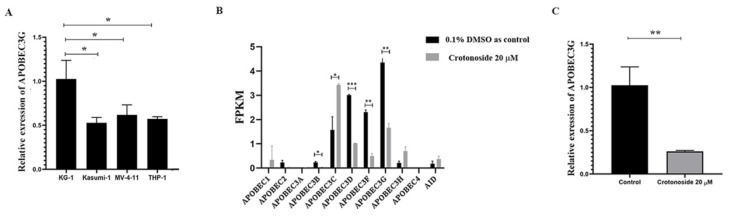
The mRNA expressions of the APOBEC3 family were evaluated in tested cells: (**A**) The relative endogenous mRNA expressions of APOBEC3G in AML cell lines were detected by qPCR. * *p* < 0.05 vs. KG-1. (**B**) The expression and changes in all the members of APOBEC family were detected by mRNA sequencing in KG-1 cell treated with 20 μM crotonoside or with 0.1% DMSO as control for 48 h.* *p* < 0.05, ** *p* < 0.01, *** *p* < 0.001 vs. control group. (**C**) KG-1 cells were treated with 20 μM crotonoside or 0.1% DMSO as control for 48 h, qPCR detected the efficiency of crotonoside on inhibiting the high expression of APOBEC3G compared with control, ** *p* < 0.01 vs. control group. All the data are presented as means ± SD from three independent experiments. FPKM: fragments per kilo base per million.

**Figure 9 molecules-27-05804-f009:**
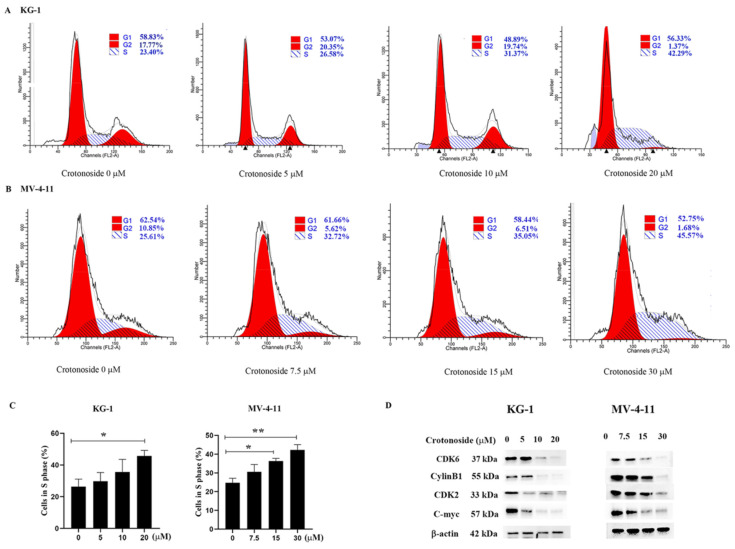
Effect of crotonoside on cell cycle progression of KG-1 and MV4-11 cells: (**A**) KG-1 and (**B**) MV-4-11 cells were treated with different concentrations of crotonoside for 48 h, and 0.1% DMSO (0 μM) as control, then the cells were harvested and cell cycle was analyzed by flow cytometry; (**C**) Statistical summary of KG-1 and MV-4-11 cells in S phase based on cell cycle analysis, * *p* < 0.05, ** *p* < 0.01 vs. 0 μM. (**D**) Effect of crotonoside on cell cycle-related proteins. After treatment with different concentrations of crotonoside for 48 h, whole-cell lysates were analyzed by Western blot. All the data are presented as means ± SD from three independent experiments.

**Figure 10 molecules-27-05804-f010:**
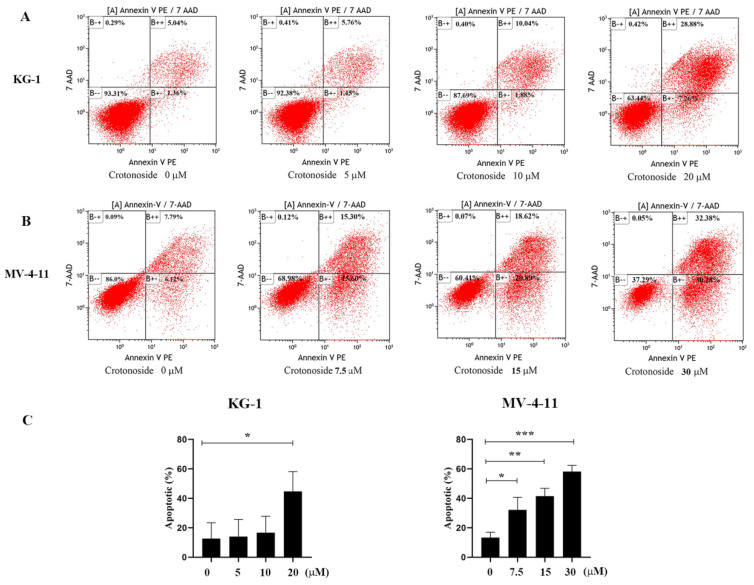
Effects of different concentrations of crotonoside on AML cell apoptosis: (**A**) KG-1 cells and (**B**) MV-4-11 cells were exposed to crotonoside at different concentrations for 48 h, and 0.1% DMSO (0 μM) as control, then cells were harvested and stained with annexin V-PE/7-AAD for flow cytometry analysis Dual staining for annexin V-PE and 7-ADD permitted the discrimination among live cells (B − −), early apoptotic cells (B + −), late apoptotic cells (B + +), and necrotic cells (B − +); (**C**) Statistical summary of apoptosis based on flow cytometry dot plots graphs, * *p* < 0.05, ** *p* < 0.01, *** *p* < 0.001 vs. 0 μM. All the data are presented as means ± SD from three independent experiments.

**Table 1 molecules-27-05804-t001:** Association between APOBEC3G expression and the clinic pathological characteristics in AML patients.

Characteristic	Low Expression of APOBEC3G	High Expression of APOBEC3G	*p* Value
n	75	76	
Gender, n (%)			1.000
Female	34 (22.5%)	34 (22.5%)	
Male	41 (27.2%)	42 (27.8%)	
Race, n (%)			0.317
Asian	1 (0.7%)	0 (0%)	
Black or African American	8 (5.4%)	5 (3.4%)	
White	64 (43%)	71 (47.7%)	
Age, n (%)			0.554
≤60	46 (30.5%)	42 (27.8%)	
>60	29 (19.2%)	34 (22.5%)	
WBC count (×10^9^/L), n (%)			0.255
≤20	34 (22.7%)	43 (28.7%)	
>20	40 (26.7%)	33 (22%)	
BM blasts (%), n (%)			0.369
≤20	33 (21.9%)	27 (17.9%)	
>20	42 (27.8%)	49 (32.5%)	
PB blasts (%), n (%)			0.932
≤70	35 (23.2%)	37 (24.5%)	
>70	40 (26.5%)	39 (25.8%)	
FAB classifications, n (%)			<0.001
M0	1 (0.7%)	14 (9.3%)	
M1	17 (11.3%)	18 (12%)	
M2	19 (12.7%)	19 (12.7%)	
M3	15 (10%)	0 (0%)	
M4	16 (10.7%)	13 (8.7%)	
M5	6 (4%)	9 (6%)	
M6	0 (0%)	2 (1.3%)	
M7	0 (0%)	1 (0.7%)	
Cytogenetic risk, n (%)			0.210
Favorable	19 (12.8%)	12 (8.1%)	
Intermediate & Poor	55 (36.9%)	63 (42.3%)	
FLT3 mutation, n (%)			0.137
Negative	46 (31.3%)	56 (38.1%)	
Positive	27 (18.4%)	18 (12.2%)	
NPM1 mutation, n (%)			0.018
Negative	52 (34.7%)	65 (43.3%)	
Positive	23 (15.3%)	10 (6.7%)	
CEBPA mutation, n (%)			0.579
Negative	70 (46.4%)	68 (45.0%)	
Positive	5 (3.3%)	8 (5.3%)	
DNMTA mutation, n (%)			0.598
Negative	59 (39.1%)	56 (37.1%)	
Positive	16 (10.6%)	20 (13.2%)	

**Table 2 molecules-27-05804-t002:** Univariate and multivariate Cox’s regression analysis of factors associated with OS in AML.

Characteristics	Univariate Analysis		Multivariate Analysis	
	Hazard Ratio (95% CI)	*p* Value	Hazard Ratio (95% CI)	*p* Value
Age (>60 vs. ≤60)	3.333 (2.164–5.134)	<0.001	3.057 (1.915–4.880)	< 0.001
Gender (male vs. female)	1.030 (0.674–1.572)	0.892	0.865 (0.527–1.419)	
WBC count (×10^9^/L) (>20 vs. ≤20)	1.161 (0.760–1.772)	0.490	0.838 (0.669–2.058)	
BM blasts (%) (>20 vs. ≤20)	1.165 (0.758–1.790)	0.486	1.001 (0.573–1.748)	
PB blasts (%) (>70 vs. ≤70)	1.230 (0.806–1.878)	0.338	1.195 (0.634–2.255)	
Cytogenetic risk (Intermediate & Poor vs. Favorable)	3.209 (1.650–6.242)	< 0.001	2.189 (0.975–4.915)	0.058
FAB classifications (M0 vs. M3)	3.386 (1.036–11.059)	0.043	0.700 (0.163–3.003)	0.632
FAB classifications (M1 vs. M3)	3.738 (1.264–11.056)	0.017	1.639 (0.453–5.929	0.451
FAB classifications (M2 vs. M3)	3.574 (1.219–10.477)	0.020	1.091 (0.303–3.929)	0.895
FAB classifications (M4 vs. M3)	3.979 (1.346–11.764)	0.013	1.460 (0.413–5.159)	0.557
FAB classifications (M5–M7 vs. M3)	6.615 (2.117–20.666)	< 0.001	2.263 (0.560–9.147)	0.252
FLT3 mutation	1.271 (0.801–2.016)	0.309	2.206 (1.233–3.947)	
NPM1 mutation	1.137 (0.706–1.832)	0.596	0.545 (0.285–1.041)	
APOBEC3G	1.893 (1.230–2.914)	0.004	1.917 (1.175–3.126)	0.009

## Data Availability

The datasets analyzed for this study can be found in the National Cancer Institute (NCI) TCGA cancers (TCGA-LAML) https://portal.gdc.cancer.gov/; accessed on: 10 July 2021, GTEx (normal tissues) https://gtexportal.org/home/datasets; accessed on: 10 July 2021, and Gene Expression Omnibus (GSE63270) https://www.ncbi.nlm.nih.gov/geo; accessed on: 10 July 2021. All data that support the findings of this study are available from the corresponding author upon reasonable request.

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
