# Peer review of "The Identification of APOBEC3G as a Potential Prognostic Biomarker in Acute Myeloid Leukemia and a Possible Drug Target for Crotonoside"

_molecules, 2022, doi:10.3390/molecules27185804_

Round 1

Reviewer 1 Report (New Reviewer)

The manuscript is well written and contains significant results. The main drawback is the lack of controls. There are no results of APOBEC3G levels in the healthy cohort or in normal cells. In vitro assays should be carried out in normal cell lines. These data are essential to properly evaluate the APOBE3G levels in AML.

Other comments to consider:

11)     The CCK-8 assay determines cell viability; however, the mansucript makes reference to cell proliferation in the abstract, introduction (reference number 27 evaluated cell viability), results (2.6. and 2.8), and line 350. The difference between ‘cell viability’ and ‘proliferation’ should be taken into account for the interpretation and discussion of the results.

22)     Figure 9B includes the cell line MV-4-11. It should be mentioned in the footnotes.

33)     The protein expression of the cell cycle related protein has been determined in the MV-4-11 cell line; however, it has not been used to measure apoptosis and cell cycle. The use of this cell line should justify and explain why apoptosis and cell cycle assays have not been carried out on this cell line.

44)     In Figure 8, asterisks (significant differences) should be drawn above each data group/bar compared, and include in footnotes which group has been compared.

55)     The lack of statistics in Figure 10 (asterisks) and the conditions of control treatment should be described in the footnotes.

66)     There is no a Figure 11.

77)     The title of the footnotes should be more precise (Figure 8).

88)     Lack of information in footnotes in some figures (abreviations, meaning of colors, control treatment, duration of treatment, vehicle, statistics).

99)     Graphs are not seen clearly (the font size in the axis is so small, the chart size).

A Manuscript version with these comments and some others is attached.

Author Response

Reviewer 2 Report (New Reviewer)

This is a well-written manuscript on the role of APOBEC3G in acute myeloid leukaemia (AML) and the inhibition of this apolipoprotein by crotonoside. The experimental design is rigorous, and the manuscript has been organised easy for readers to follow and catch the information. Although crotonoside displayed moderate activities against AML cells (IC50 = 60.69-20.77 µM), understanding its mechanism of action is essential for developing new drugs. Therefore, I believe the manuscript could be of great interest in medicinal chemistry, especially that related to the development of anticancer compounds.

Author Response

Thank you very much for your review of our articles and your approve of our work.

Round 2

Reviewer 1 Report (New Reviewer)

Dear authors, 

Thank you for all efforts and changes that have improved the quality of the manuscript. Some comments about the revision version of the manuscript: 

- The term 'cell proliferation' to refer to the results obtained by the CCK-8 assay (cell viability assay) is not totally correct. A compound can decrease the viability, or a compound can be cytotoxic, but not inhibit the cell viability.

-Other little considerations, such as to indicate which value are being compared to obtain an p<0.0001 in figure 8B. Or the term “proporation” (point 2.8).

Author Response

 Response to the reviewer’s comments:

1.The term 'cell proliferation' to refer to the results obtained by the CCK-8 assay (cell viability assay) is not totally correct. A compound can decrease the viability, or a compound can be cytotoxic, but not inhibit the cell viability.

Response: Thank you very much for your professional comments. By reviewing the literature, we have deleted the description of cell proliferation in the article. From our point of view, CCK-8 is considered to be used for detection of cell viability .Based on the results of CCK-8, crotonoside can inhibit viability of AML in a dose-depended manner. Combined with the results of cell cycle analysis, and we discovered the S phase arrest in cell cycle. Therefore we deduced that crotonoside prevented the cell cycle, and resulted in cell viability decreased. In addition, according to the results of apoptosis experiments, we believe that crotonoside has certain cytotoxicity and can induce apoptosis of cells. We don’t sure whether our opinion is consist with the reviewer. If there is something wrong, we hope to get corrections from the reviewer.

2. Other little considerations, such as to indicate which value are being compared to obtain an p<0.0001 in figure 8B. Or the term “proporation” (point 2.8).

Response: Thank you for pointing out the problem. I'm very sorry for the mistakes in the figures and footnotes. As the reviewer’s suggestion, we had indicated the values which are being compared in the figures and colored them in red.. The term “proportion” was a spelling mistake, and it had been corrected in part 2.8, and colored them in red.

This manuscript is a resubmission of an earlier submission. The following is a list of the peer review reports and author responses from that submission.

Round 1

Reviewer 1 Report

The manuscript by Ma C. and co-workers attempts to identify APOBEC3G as a new potential prognostic biomarker in acute myeloid leukemia and a target of crotonoside. The findings were based on a bioinformatic analysis and in vitro assays with different AML cell lines. There are several pitfalls with the bioinformatic and experimental strategies as follows:

- The cohorts from TCGA database have to be properly identified. The control cohort is not identified and its match with the LAML cohort is not known. Regarding the LAML cohort, it is not possible to understand by the manuscript if the database is from bone marrow cells, leukemic blasts, peripheral blood cells, etc.

- On table 1, the FAB classification is missing for one low expression AML patient. In addition, FAB classification has been substituted by WHO classification. Although NPM1 information is given, information on CEBPA mutations, among other, is not available, which jeopardizes prognostic value and the identification of APOBEC3G as a prognostic marker.

- AUC values for ROC curves were 0.610, 0.646, 0.716, which suggests that only for the three-year the AUC value is acceptable (above 0.7), which means that APOBEC3G has no discrimination for prognosis, jeopardizing the main conclusions of authors.

- The docking simulation misses the possible binding of crotonoside to other APOBEC factors, which might be important in some scenarios. In addition, important details about this simulation are also missing.

- It is not clear the criteria used to select the cell lines. It is also not understandable why cells with low expression of APOBEC3G were not used. This would be the perfect control for the experiments and to prove the really value of crotonoside. In addition, KG-1 cells are not M2 cells.

- The conclusions on the regulated cell death induced by crotonoside in KG1 cells are not correct. Cells positive for 7AAD and Annexin V could be late necrotic cells or necrotic cells. More parameters would have to be determined to prove that the regulated cell death is apoptosis.

- Discussion has to be shortened and this section has to be reformulated because conclusions are not supported by evidences. The results do not support that APOBEC3G is an independent prognosis biomarker. Line 321 mentions a RNA sequencing of APOBEC family that was not done, among other.

- The terminology “normal persons” must be avoided. As suggestions control cohorts or healthy individuals should be used.

- Figure 8 misses statistical analysis.

- Figure 9, the different plots and graphs are not identified. The two graphs on the bottom are redundant, the flow cytometry histograms are not properly referred.

- Numerous typing errors like point and comas together, different nomenclature for the same cells, repetitions of words, title of 2.4 section, etc are found all over the manuscript. 

Reviewer 2 Report

See attached PDF for my comments
